# Metabolite Profiling of the Resurrection Grass *Eragrostis nindensis* During Desiccation and Recovery

**DOI:** 10.3390/plants14040531

**Published:** 2025-02-09

**Authors:** Erikan Baluku, Llewelyn van der Pas, Henk W. M. Hilhorst, Jill M. Farrant

**Affiliations:** Department of Molecular and Cell Biology, University of Cape Town, Cape Town 7700, South Africa; erikanbaluku@gmail.com (E.B.); llewelyn.vanderpas@uct.ac.za (L.v.d.P.); henk.hilhorst@uct.ac.za (H.W.M.H.)

**Keywords:** *Eragrostis nindensis*, gas chromatography, metabolites, resurrection grass, roots, senescence

## Abstract

Resurrection plants employ unique metabolic mechanisms to protect themselves against damage caused by desiccation. This study aimed to identify metabolites, using gas chromatography–mass spectrometry, which were differentially abundant in *Eragrostis nindensis* at different stages of dehydration and rehydration in leaves which are destined to senesce on desiccation termed “senescent tissue” (ST) and those which remain desiccation-tolerant during water deficit and are termed “non-senescent tissue” (NST). Furthermore, the study compared the shoot and root systems during extreme water deficit and recovery therefrom to unravel similarities and differences at the whole plant level in overcoming desiccation. Shoot metabolomics data showed differentially abundant metabolites in NST, including raffinose, sucrose, glutamic acid, aspartic acid, proline, alpha-ketoglutaric acid, and allantoin, which act as major drivers for plant desiccation tolerance and aid the plant post-rehydration. The metabolites which accumulated in the ST-indicated initiation of programmed cell death (PCD) leading to senescence. The roots accumulated fewer metabolites than the shoots, some exclusive to the root tissues with functions such as osmoprotection, reactive oxygen species quenching, and signaling, and thus proposed to minimize damage in leaf tissues during dehydration and desiccation. Collectively, this work gives further insight into the whole plant responses of *E. nindensis* to extreme dehydration conditions and could serve as a model for future improvements of drought sensitive crops.

## 1. Introduction

Water is essential for all life reactions on earth, and the loss of just a small amount of water results in the death of most. Plants have evolved numerous mechanisms to avoid, resist, or escape water deficit under drought conditions. However, viability is lost in the majority of terrestrial plants if cellular water content drops below a critical value, ranging from a loss of 10–30% cellular water in most annual crops to a maximum of 59% in the desert shrub *Larrea tridentata* [1,2,3]. One of the more rare and extreme mechanisms of drought tolerance is vegetative desiccation tolerance (VDT), where vegetative tissues (like orthodox seeds) are able to survive the loss of up to 95% cellular water (i.e., survive drying to a leaf water potential of approximately −100 megapascals (MPa)) for extended periods and regain full metabolic activity in most of their tissues within hours of rehydration [4]. In light of global warming, where the most crippling effect is the occurrence of increased intensity and extended droughts [5], food security, built on a small number of annuals bred in the green revolution for increased yield, but with compromise to natural resilience such as water deficit, is at risk. Thus, researchers have increasingly turned to understanding the mechanisms of VDT with the ultimate aim of improving crop tolerance to water deficit stress [6,7,8,9].

Over the past two decades, there has been increased omics-based research on desiccation-tolerant angiosperms (commonly referred to as resurrection plants), from which the following broad conclusions can be drawn: firstly, VDT evolved from the reprogramming of the genetic networks controlling desiccation tolerance in their seeds [4,10]. These form a core of seemingly common traits, but secondly, considerable ancillary mechanisms, such as osmoprotection, antioxidant activity, and the formation of a protective protein (heat shock protein and late embryogenesis abundant proteins), have been independently evolved in different families, genera, and even species [6,7,8]. While there is evidence for the convergent evolution of DT [11], there is no “blueprint” for VDT, and it is important to use the correct model resurrection plant to improve the crop. VDT also requires extensive, multifaceted characterization of the precise mechanisms employed by the model species.

*Eragrostis nindensis* is a desiccation-tolerant grass [12,13], one of only 40 species in the Poaceae described as employing VDT [14]. It is the only resurrection plant that is a close relative to the orphan crop *Eragrostis tef* (teff). To date, physiological aspects associated with dehydration, desiccation, and recovery of *E. nindensis* have been characterized [12,15,16], its genome has been sequenced, and transcriptional changes associated with leaf desiccation have been reported [13].

Physiological studies have shown that *E. nindensis* is poikilochlorophyllous, a strategy in which chlorophyll is broken down and thylakoids disassembled during dehydration to avoid the accumulation of photosynthetic reactive oxygen species (ROS). Upon rehydration, these are rapidly reassembled and regain metabolic functioning [17]. In the case of *E. nindensis*, this regreening starts approximately 24 h post-rehydration, but only reaches pre-dehydration levels after more than 72 h post-rehydration. This is in contrast to the strategy of homoiochlorophylly, adopted by some grasses and most dicotyledons, in which the entire photosystem is retained but protected via anatomical and biochemical “sun shading” mechanisms, the protection of the thylakoids, and a robust antioxidant response [4]. Physiological studies have also revealed that fully mature (older) leaves senesce after desiccation, but are more phenotypically evident on rehydration [17].

It is clear that resurrection plants are able to suppress drought-induced senescence in the bulk of their tissues [18], but the mechanisms whereby this happens have only been explored in a few species. The most comprehensive study to date has been on the grass *Tripogon lolilifolia*, where it was demonstrated that in leaf tissues, programmed cell death is suppressed by trehalose-regulated autophagic removal of toxins [19]. In roots, there is no evidence of autophagy, despite high trehalose levels, and it is postulated that the resources in the roots provide sufficient energy to cope with stress, thus not requiring autophagy [20]. Studies on leaves of the resurrection plant *Xerophyta schlechteri* have not entirely supported the trehalose-related autophagy phenomenon but pointed to a potent antioxidant regulation [21].

Of relevance, older tissues of resurrection plants do senesce on desiccation, and we have postulated that in understanding the molecular aspects of such tissue during desiccation and recovery, in comparison to surviving tissue, we are better poised to identify regulatory mechanisms, due to their failure in tissue “predisposed” to senesce on desiccation [21]. We propose that senescence is induced due to age-related failure of “tolerance” mechanisms in such tissues. With this rationale, and to further characterize mechanisms of desiccation tolerance in *E. nindensis*, this study undertook to investigate metabolomic changes in young leaves that survive desiccation and mature leaves destined to senesce on drying, during dehydration, desiccation, and rehydration. Furthermore, metabolomic changes in root tissues, which do not senesce on desiccation, were investigated.

## 2. Results

### 2.1. Change in Relative Water Content and Leaf Morphology of NST, ST, and Root

Two phases of dehydration were apparent before the plant reached the air-dry state (<5%), and this was seen by a decrease in relative water content (RWC) and a change in leaf color (Figure 1). An early response was evident when leaves dropped from full turgor to 55% RWC, where leaves remained green and uncurled, and a late response below this RWC, when leaves started curling and with apparent loss of chlorophyll (Figure 1B). This was evident in both the NST and ST, a characteristic signature of poikilochlorophylly. Roots, by contrast, rapidly lost water during the first five days of withholding water, reaching 30% RWC at this stage, and this gradually reduced until the air-dry state at 15 days after withholding water. Upon rehydration, both roots and NST gained hydration; this initially happened more rapidly in roots, with little to no rehydration occurring in ST. 

### 2.2. Comparative Metabolic Analysis Between NST and ST During Dehydration and Rehydration

A total of 97 metabolites were identified, of which 83 were differentially abundant (*p* < 0.05) between NST and ST, consisting of sugars, sugar alcohols, organic acids, amino acids, phytohormones, fatty acids, and other metabolites (Appendix A). Metabolic changes between ST and NST across each dehydration stage were indicated by principal component analysis (PCA) (Appendix A). Furthermore, the fold change (log2FC) of all the significant results are presented in the Appendix A.

Among the amino acids, there was an early decrease in aspartic and L-glutamic acid, with these accumulating at the air-dry state in NST, unlike the ST (Figure 2). The accumulation of the latter is consistent with its role as the primary nitrogen donor for synthesizing other amino acids required post-rehydration, as this attests to the accumulation of almost all amino acids during rehydration in NST with a subsequent decrease in glutamic acid (Figure 2). However, the accumulation of L-leucine, DL-isoleucine, and L-valine was more pronounced in ST, with lower abundance observed in NST at the air-dry state and post-rehydration. Their accumulation could be related to fast protein turnover or stress-induced protein degradation, and similarly, high levels have been reported in *Sporobolus stapfianus* [22] and *X. schlechteri* [23] during drought stress.

#### 2.2.1. Organic Acids

Concerning organic acids, alpha-ketoglutaric acid was found in greater abundance in the air-dry state in NST when compared to other organic acids (Figure 3), which could be acting as a carbon skeleton for nitrogen assimilation to aid in the survival of the NST. Contrary to this, the ST exhibited elevations in cinnamic acid, which are reported to cause cell wall lignification [24], which suggests ST is geared towards senescence. The decline in tricarboxylic acid (TCA) cycle intermediates (fumaric-, pyruvic-, and D-malic acid) during dehydration is likely to play a role in the decline of respiration and the shift in metabolism towards the synthesis of metabolites crucial for protection during desiccation, as similar findings have been reported in other resurrection plants [22,23]. A high abundance of phosphorous, mucic, lactic, pyruvate, and succinic acid was observed 48 h post-rehydration, more pronounced in NST, an indicator of resumption of respiration and normal metabolism in the NST, contrary to the ST.

#### 2.2.2. Sugars and Sugar Alcohols

Sugars (glucose and fructose) and some sugar alcohols (scyllo-inositol, myoinositol, and D-sorbitol) increased predominantly in NST during early dehydration, which is likely to indicate their osmoregulatory and priming role in preparing the plant for subsequent water loss. There was a notable accumulation of sucrose and raffinose in the air-dry state (Figure 4A), the latter accumulating exclusively in NST, which indicates its role in desiccation, as reported in other resurrection plants [21,22,23]. However, raffinose levels in NST remained high at 24 h post-rehydration and only decreased at 48 h with increasing levels of maltose, cellobiose, galactose, fructose, and glucose, indicating a significant shift in metabolism in NST, unlike the ST which showed low levels of sugars post-rehydration. There was also a substantial decrease in myo-inositol, the precursor metabolite for raffinose synthesis, in the air-dry state, as depicted in Figure 4A, followed by an increase in glycerol and myo-inositol predominantly in NST post-rehydration. Indole-3-acetic acid (IAA) and 4-amino butanoic acid (GABA) acid accumulated post-rehydration in both tissues, peaking at 48 h, albeit more substantially in the NST (Figure 4B). Allantoin accumulated in NST being maximal in the air-dry state. Accumulation of this metabolite has been linked to the up-regulation of genes related to ROS scavenging [25].

### 2.3. Comparative Pathways Analysis for Differentially Abundant Metabolites in the Leaf Tissue

During dehydration, metabolites involved in the TCA cycle, glyoxylate and dicarboxylate metabolism, ascorbate, and aldarate metabolism, were less abundant exclusively in the NST, as indicated in Figure 5A. Glycolysis and pyruvate metabolism were reactivated post-rehydration in the NST (Figure 5B), indicating the resumption of metabolic activity. Starch and sucrose metabolism were common to all dehydration stages in the NST, consistent with high levels of sucrose and raffinose. However, none of the metabolic pathways were exclusively associated with metabolites accumulating in ST during rehydration. Metabolites related to central carbon metabolism, namely glycolysis/gluconeogenesis, were exclusively less abundant in NST during dehydration.

### 2.4. Comparison of Differentially Abundant Metabolites in Shoot and Root System

There was a high abundance of raffinose, sucrose, glycerol, maltose, succinic acid, glucose, fructose, glutamic acid, proline, aspartic acid, and tryptophan in the NST with decreased abundances in the roots during dehydration and rehydration stages (Figure 6). However, the roots showed a high abundance of allantoin, tryptophan, and trehalose in the air-dry state, implying that these metabolites could be involved in root desiccation tolerance. In contrast, levoglucosan accumulated at 24 and 48 h post-rehydration. The slight increase in trehalose in the roots at the air-dry state indicates a role in the desiccation tolerance of the root system. Most of the amino acids and sugars were predominantly low upon rehydration in roots, as depicted in Figure 6. Organic acids were less abundant during dehydration, and only fumaric, succinic, and L-ascorbic acids were present at 48 h rehydration. Some of the metabolites were exclusive to roots and are shown in Appendix A.

## 3. Discussion

In the present study, we investigated the metabolomic changes between the desiccation-sensitive (ST) and desiccation-tolerant (NST) leaves of *E. nindensis*. We also coupled changes in root and shoot to further investigate the metabolomic drivers of desiccation tolerance and survival in the NST compared to the subsequent senescence of the ST. The distinction between the NST and ST based on visual cues during dehydration cannot distinguish the two tissue types. However, our data indicated that significant metabolomic reprogramming occurred in the NST, which was absent in the ST.

### 3.1. Relative Water Content and Leaf Morphological Changes

While no differences in the rate of water loss were observed between the NST and ST, only the NST recovered water content and metabolic activity upon rehydration (Figure 1A, Figure 2, Figure 3 and Figure 4). However, morphologically, the NST showed controlled leaf rolling at 55–45% RWC, whereas the ST did not (Figure 1B). Coordinated leaf folding is a typical response in resurrection plants, and it has been linked to lower transpiration rates, which aids in temperature regulation and provides photoprotection [21]. The lack thereof in the ST likely contributes to the lack of survival following desiccation and, ultimately, commitment to senescence, clearly noted by the lack of recovery in these leaves (Figure 1B). While senescence in leaves has been attributed to several factors and is still not clearly defined for most resurrection plants, including *E. nindensis*, a study by Vander Willigen, Pammenter, Mundree, and Farrant [15] comparing the desiccation-sensitive species *E. curvula* with *E. nindensis* showed that a senescent phenotype is more evident in the outer (more mature) leaves of *E. nindensis*, similar to that observed in the *E. curvula*. This is further corroborated by the observation that low levels of antioxidant-related metabolites were present in the ST during dehydration (Figure 2), a phenomenon reported to be a driver of senescence (and the reverse being an inhibitor of senescence) in the poikilochlorophyllous resurrection plant *X. schlechteri* [21].

The initial rapid loss of water in roots could be due to the “cohesion-tension” hypothesis, which proposes that one of the ways water moves across plants from roots to leaves is caused by transpiration to facilitate stomata opening, allowing a gaseous exchange for photosynthesis [26,27]. Indeed, the study by Ginbot and Farrant [12] on *E. nindensis* showed an apparent decline in the transpiration rate after 3 days of dehydration, which suggests a substantial decrease in root RWC at this stage.

### 3.2. Sugars and Sugar Alcohols as Osmoprotectants and Vitrification Agents in NST

Vitrification, the formation of a glassy state, is a critical mechanism that resurrection plants and seeds use to preserve their cellular structures during the later stages of desiccation [28,29]. In the NST, particularly in the late stages of dehydration, there is an increase in the levels of raffinose and sucrose. This accumulation supports the idea that these sugars facilitate dehydration and desiccation tolerance [22,23,30,31]. Sugar alcohols such as galactinol and myo-inositol have been reported to serve as osmoprotectants and antioxidants [32]. However, their reduction in *E. nindensis* suggests that their role in *E. nindensis* may be limited to raffinose family oligosaccharides (RFOs) synthesis.

Furthermore, raffinose has been identified as an inhibitor of sucrose crystallization [33], implying that its accumulation may also help stabilize sucrose within the glassy matrix. This stabilization contributes to vitrification, enhancing the plant’s ability to withstand desiccation. In contrast, the absence of raffinose accumulation during drying in the ST indicates that effective vitrification was not established, which may lead to PCD. This observation further suggests that desiccation tolerance may be “lost” as the leaves mature. Given the perennial nature of *E. nindensis*, leaves are regularly shed during growth. As such, the apparent “loss” of desiccation tolerance in the ST might be related to age-dependent leaf shedding.

During initial rehydration, the decrease in sucrose correlates with the accumulation of simple sugars such as fructose and glucose, and these molecules serve as carbon sources for the regeneration of ATP through glycolysis in the NST, with an increase in sucrose in this tissue at one week post-rehydration implying complete restoration to full metabolic activity. The low levels of raffinose at 48 h post-rehydration further suggest that its catabolism could contribute to the formation of sucrose. This correlates with a previous study by Radermacher, du Toit, and Farrant [21], who reported decreased levels of sucrose and raffinose post-rehydration in the NST of *X. schlechteri*. These metabolites, by comparison, are less abundant in ST (Figure 5B), suggesting failure to restore full metabolic activity. The elevated glycerol levels at 48 h post-rehydration in NST suggest a mechanism to prevent subsequent water loss through lowering the surface tension of water and exhibiting a strong water binding activity, a process reported by Weng et al. [34]. On the contrary, the low levels seen in ST further imply the failure to overcome the continuous loss of water, rendering ST desiccation sensitive upon late dehydration.

Although a study by Norwood et al. [35] reported high levels of sucrose in the roots of *Craterostigma plantagineum*, we observed low levels of sucrose, galactinol, and sorbitol (Figure 6), suggesting that *E. nindensis* could be draining its carbohydrate reserves in the roots to the shoot system to aid in the survival of the NST, though not much has been reported to validate this hypothesis. However, there is a predominant increase in trehalose levels in the roots at the air-dry state. Trehalose is reported to play a role in abiotic stress and regulating autophagy in the roots of *T. loliiformis* [19]. Along this line, it is tempting to speculate that trehalose plays a different role than sucrose in the roots of *E. nindensis*, suggesting, perhaps, a unique survival strategy for roots to survive desiccation. A study conducted by Asami, Rupasinghe, Moghaddam, Njaci, Roessner, Mundree, and Williams [20] reported that the accumulation of sucrose and trehalose-6-phosphate in the roots of *T. loliiformis* assisted the plant in the survival of desiccation. This clearly supports the idea that trehalose accumulation could likely maintain continuous metabolic activity during desiccation to facilitate water and nutrient uptake during dehydration and post-recovery.

### 3.3. Regulation of Photosynthesis and TCA During Dehydration and Rehydration in NST

Cessation of photosynthesis and shutdown of metabolism are well-reported strategies poikilochlorophyllous resurrection plants employ in response to dehydration and desiccation as a strategy to avoid ROS accumulation [36,37]. These mechanisms are further reported in *Xerophyta* species [21,23,36] and *C. plantagineum* [8]. In agreement with this, most of the TCA intermediates decreased predominantly in the NST (Figure 3). However, the increasing levels of ketoglutarate in the air-dry state is likely to be attributed to its role in the GABA shunt, which is a metabolic process to replenish succinic semialdehyde (SSA) into TCA for the cellular metabolic process. A study by Ginbot and Farrant [12] cited that respiration is critical for resurrection plants to maintain their ability to provide ATP for the repair of stress-induced damage and facilitate the acquisition of desiccation tolerance. The high level of ketoglutarate is a continuous and indirect supply of TCA intermediates facilitating the production of ATP in the NST to sustain passive cellular metabolic activity during desiccation. Ketoglutarate enhances nitrogen assimilation [38], and this could be contributing to the high levels of amino acids post-rehydration in NST, thus playing a role in the survival of the tissue. The elevated levels of GABA during initial dehydration could thus be related to a similar role. Plants mutated to synthesize SSA have shown ROS accumulation and senescence in their leaves [39]. The absence of the GABA shunt in the ST further supports the notion that these tissues commit to senescence.

During initial rehydration, there is a significant metabolic shift in TCA intermediates, predominantly in the NST. However, low levels of malic acid were evident, and this could be related to its utilization in the production of NADPH, a reducing agent utilized in photosynthesis and required for several antioxidant enzymes involved in glutathione and glutamate metabolism. The increase in pyruvic acid levels at 48 h post-rehydration implies that *E. nindensis* could be utilizing the methylerythritol phosphate pathway to synthesize carotenoids, which are important in photoprotection. High levels of pyruvic acid were reported in *E. nindensis* (Vander Willigen, Pammenter, Mundree, and Farrant [17]) and in *C. wilmsii* [36]. In this case, poikilochlorophyllous and homoiochlorophyllous plants utilize a similar mechanism to mask chlorophyll from excessive radiation during recovery, though this strategy is more evident in the latter. The roots could be utilizing other strategies as discussed since no changes were noticed. The low levels in the ST, by comparison, yet again support the idea of failure to resume metabolic activity.

### 3.4. Antioxidant Activity Mechanisms During Dehydration and Rehydration

There was a high abundance of antioxidant-associated metabolites such as ascorbic acid exclusively in the NST at early dehydration and amino acids such as proline, tryptophan, and phenylalanine at the air-dry state in both the ST and NST. The accumulation of ascorbic acid in the NST at early dehydration (Figure 3) is associated with its role in the antioxidant network formation in plants, and most importantly, the uniqueness of forming five distinct isoforms makes it an efficient ROS scavenger in both the cytosol and chloroplast [40]. Furthermore, ascorbic acid is also used as a reducing agent to reduce hydrogen peroxide (H_2_O_2_) to water (H_2_O), as a ROS scavenger [41,42,43], and Fe^3+^ to increase its uptake by plants [44]. Iron reduction facilitates cellular activities and reduces PCD facilitated by ferroptosis (iron-induced lipid peroxidation) [45] in plant cells. Furthermore, the high abundance of putrescine, methionine, and ornithine in the NST indicates a robust antioxidant activity, unlike the ST, which accumulated proline only, which is also reported in sensitive species, thus suggesting once more that ST displays desiccation sensitivity during late dehydration [46]. However, proline is also reported to accumulate under abiotic stress [47] and preserve the cellular components to facilitate cell function during desiccation [48]. The accumulation of proline in our study agrees with the reported levels in *E. brachyphylla*, *X. humilis* [30], *S. stapfianus* [49], *Selaginella bryopteris* [50], and *S*. *tamariscina* [51], suggesting a similar role in *E. nindensis*. Since the accumulation of most antioxidants was more pronounced in the NST than the ST, we speculate that the ST is geared towards senescence. Besides antioxidant activity, amino acids also play a role in nitrogen metabolism [52]; the high abundance of glutamic acid in the dry state, predominantly in the NST, relates to this role [53]. The accumulation of glutamic acid and aspartic acid implies an additional role in the glutamine oxoglutarate aminotransferase (GOGAT) pathway to synthesize amino acids which serve as antioxidants and osmolytes. Therefore, the accumulation of these two amino acids and ketoglutarate in the air-dry state implies that the GOGAT pathway is operational in the NST but not in the ST, rendering the latter more sensitive to desiccation.

The low levels of glutamate at 24 h post-rehydration, followed by a significant increase at 48 h agree with previous findings, which reported a decrease in glutamine levels at 24 h post-rehydration in *Haberlea rhodopensis* [54]. The initial decrease suggests that glutamate is involved in the synthesis of amino acids, seen to accumulate at 48 h post-rehydration predominantly in the NST (Figure 2), and plays several roles such as regulating root and shoot architecture, acting as signaling molecules, and regulating flowering in plants [55]. The low levels of glutamic acid shown in the roots 48 h post-rehydration (Figure 6) suggest that amino acids are likely to be translocated from source (roots) to sink (leaves) sites [56] since nitrogen metabolic processes take place in leaves during recovery. Equally, the accumulation of branched-chain amino acids (BCAAs) in the air-dry state in the ST raises more speculation as to whether these amino acids are also translocated and utilized in nitrogen recycling to generate large amounts of ATP [57] to aid in the survival of the NST. However, this might not be conclusive, and further investigations are needed to validate this hypothesis, despite similar observations reported in a study by Martinelli, Whittaker, Bochicchio, Vazzana, Suzuki, and Masclaux-Daubresse [49], where the increase in BCCAs upon dehydration in older leaves of *S. stapfianus* aligns with the pattern observed in the ST.

The accumulation of GABA during early dehydration (Figure 4) correlates with findings by Hasan et al. [58] who reported that plants subjected to abiotic stress generate high levels of GABA. GABA, both as a signaling molecule [59] and playing a role in the production of secondary metabolites, naturally intersects most of the metabolic pathways that involve nitrogen and carbon metabolism [60,61], thus playing multiple roles in response to desiccation [58]. We have limited knowledge of the accumulation of other metabolites, such as palmitic acid and salicylic acid, in NST upon rehydration. However, their accumulation in ST during desiccation relates to senescence since salicylic acid and ethylene accumulation play prominent roles in senescence [62]. A study by Newman [63] also reported the accumulation of fatty acids in senescent chloroplasts, which renders more credence to their accumulation in ST during desiccation.

Allantoin is a compound which is involved in nitrogen metabolism. Its increase in the roots in the dry state is consistent with the levels reported in the roots of a drought-tolerant genotype of rice [64] and bread wheat exposed to water and nitrogen limitation [65]. Again, these outcomes are comparable to those seen in *S. stapfianus*, where allantoin accumulation increased when RWC dropped to 40% or less [22], suggesting that allantoin is likely to play a comparable function in the roots of *E. nindensis* by increasing antioxidant activity as well as acting as a signaling molecule [66].

### 3.5. Comparison of Metabolic Pathway Processes in NST and ST

Upon further comparisons of the NST and ST, the enrichment of starch and sucrose metabolism exclusively in the NST during dehydration (Figure 5) further supports the hypothesis that the NST is geared towards quiescence while the ST is towards senescence. On the contrary, the ST showed an enrichment of valine, leucine, and isoleucine degradation correlating with the accumulation of BCAAs. Similar levels of BCAAs have been reported in older leaves of *S. stapfianus* [67] and *X. schlechteri* [23], suggesting that stress-induced protein degradation may be associated with the increase in amino acids. Plants may also selectively repurpose these BCCAs for the production of essential amino acids as well as alternative sources of carbon [68]. For instance, high levels of BCCAs in the NST during rehydration could be responsible for enhancing protein biosynthesis required during recovery, whereas the enrichment in the ST relates to the high protein turnover due to ubiquitin-mediated proteolysis [69], akin to being geared towards senescence.

## 4. Conclusions

This study demonstrated that the NST and ST differ in metabolite accumulation, rendering the NST desiccation tolerant and the ST desiccation sensitive. This is important for the understanding of the desiccation tolerance of *E. nindensis*. This study unraveled the key metabolites involved in the desiccation tolerance mechanism of the whole plant to mitigate the effects of desiccation, with a more significant variation in metabolites seen in leaves, mainly the NST, than in roots. The low abundance of metabolites in the roots indicates that it was less affected by desiccation, and this attests that they could be more desiccation tolerant than the leaves. However, the high accumulation of some metabolites in the roots might assist in reducing the effect of desiccation in the roots and also aid in the uptake of water and nutrients during recovery. The study confirmed that more resources are allocated to the leaves, particularly the NST, to mitigate the effects of desiccation and aid in recovery. The ST becomes desiccation sensitive at late dehydration, and this was seen by the low abundance of most metabolites involved in desiccation tolerance.

## 5. Methods and Materials

### 5.1. Plant Material and Growth Conditions

*Eragrostis nindensis* seeds were collected from plants grown in the field near Aggeneys, Northern Cape, South Africa (29°16′41.1″ S 19°00′25.4″ E) with permission from the land manager Pieter Venter. These were germinated in a mixture of potting soil, sand, and vermiculite in ratios of 2:1:1 and were maintained in a growth room under the following environmental conditions: day/night cycle: variable between 16 h in light and 8 h in dark; light intensity: variable between 250 and 500 µmol m^−2^ s^−1^; temperature: ambient temperature ranging from 25.0 °C and 27.5 °C depending on light intensity; and relative humidity: 50–60%.

### 5.2. Sampling Procedures

During the sampling, young leaf tissue was regarded as non-senescent tissue (NST) and old leaf tissue as senescent tissue (ST) based on previous findings from Willigen, Pammenter, Jaffer, Mundree, and Farrant [17]. Fully hydrated samples were taken after the last day of acclimation, where all the plants were fully watered at dusk, and sampling was performed after 15 h on the following day. Dehydration was induced by ceasing to water the soil and plants. Different dehydration sampling points were estimated based on the physical appearance of leaves during the dehydration process, especially the change in color and folding of the leaves. At each sampling event throughout the dehydration and rehydration time courses, six different plants used for leaf and root collection. Relative water content (RWC) was determined on a portion of each tissue, as outlined below, and remaining tissue individually frozen in liquid nitrogen and stored at −20 °C until the end of the time course. At that stage, samples were grouped into discreet categories namely full turgor (FT), early dehydration (ED), late dehydration (LD), air-dry (AD), 24 h (RI), 48 h (RII), and one week post-rehydration. FT ranged from 100 to 75%, ED 75–60%, LD 55–40%, and AD < 5% RWC of which five biological replicates were analyzed for metabolomics.

### 5.3. Relative Water Content Determination

The RWC of the tissues was determined using the gravimetric method outlined by Barr and Weatherley [70] using the following equation:RWC (%) = Absolute water content final/Absolute water content at full turgor × 100

### 5.4. Extraction and Derivatization

Extraction and derivatization were conducted according to Lisec et al. [71] and Valledor et al. [72], with minor modifications to the ratio of methanol to water. Ground samples were freeze-dried in a SpeedVac (Savant SpeedVac SC100, Walter Fisher Scientific, Waltham, MA, USA) and placed at 4 °C. A total of 10 mg of finely ground leaf and root tissue was placed in tubes and kept under liquid nitrogen. A methanol–water extraction solvent was used to separate metabolites based on polarity, and ribitol was added as an internal quantitative standard. Pre-cooled 1 mL of 50% (*v/v*) methanol–water extraction solvent was added to the samples, vortexed for 30 s, and then agitated for 45 min at 4 °C and 1000 rpm on an orbital shaker. The samples were then sonicated for 15 min and centrifuged for 15 min at 12,000 rpm using a benchtop centrifuge pre-cooled to 4 °C. Samples were freeze-dried for 30 min before derivatization. Derivatization was performed by the addition of 40 µL of methoxyamination reagent. Samples were shaken for 2 h at 37 °C at 1000 rpm, followed by the addition of 70 µL of pre-warmed N-methyl-N-(trimethylsilyl) trifluoroacetamide (MSTFA) and shaken for a further 30 min at 37 °C and 1000 rpm.

### 5.5. GC-MS Analysis of Metabolites

The derivatized samples were analyzed on an Agilent Model 7890A gas chromatography system equipped with 7693 autosamplers and joined with a 7000C Triple Quadrupole mass spectrophotometer (Agilent Technologies, Santa Clara, CA, USA). Using the autosampler setup in split-less mode, samples of 1 µL were injected at a pressure of 12.2 psi at 240 °C for the injection port, and a helium carrier gas flow rate of 1 mL per minute. The initial oven temperature was 80 °C for 1 min, followed by an 8 °C per minute increase to 320 °C, and the temperature was maintained for 1.5 min. The split ratio was changed to 1:29 (30 times dilution). The samples’ analytes were separated using the Agilent column J&W 122-5532G DB-5ms+DG, which has an internal diameter of 0.25 mm and a length of 30 mm. The DB-5ms+DG is a low polarity GC column fitted with a non-polar phenyl arylene stationary phase satisfactory for acquisitive signal-to-noise ratio with better sensitivity and mass spectral integrity of analytes (Agilent, Waldbronn, Germany). The real-time for the runs on the GC-MS system was preceded by a solvent delay of 6 min, making the chromatographic duration 35.5 min. The mass spectrophotometer was tuned for an ion source of 230 °C and an electron-impact (EI) mode of 70 eV, documented with a mass range of 70 to *m*/*z* 500 at 20 scans per second and a gain factor of 1.

### 5.6. Peak Identification and Data Processing

The MassHunter workstation software version B.05 GC MS/MS (Agilent, Waldbronn, Germany) was used to identify and quantify metabolites from the spectrum using acquired spectrum data and the internal standard ribitol. The spectral data were deconvoluted to identify all the metabolites in the samples. The identification and quantification of specific metabolites were based on the NIST08 MS library (National Institute of Standards and Technology, Gaithersburg, MD, USA) and comparing retention indices based on the literature from previous authors [73,74,75]. The sample that accumulated the majority of the metabolites was used to establish a master peak list, and the remaining samples were quickly aligned by analyzing the batch to the peak list formed. This created an automatic peak alignment for all metabolites in the different samples. The output was generated in Excel format, including columns for the various samples and rows for the detected metabolites. The raw data from the MassHunter B.05 GC/MS software contained the retention time, match factor signal-to-noise ratio (S/N), and peak area for all the sample analytes. The analytes were quantified by comparing their peak area with that of internal standard ribitol during the normalization process. Raw data generated from the Mass Hunter B.05 GC/MS software were exported to Microsoft Excel 2010, and the resulting data matrix was exported as comma-separated values (CSVs) file and uploaded to MetaboAnalyst 5.0 for statistical analysis.

### 5.7. Statistical Analysis

MetaboAnalyst version 5 (https://www.metaboanalyst.ca/, accessed on 9 November 2024) [76] was used for comparative analysis using One-way Analysis of Variance (ANOVA) and Fisher’s LSD post hoc analysis method with a *p*-value of 0.05. Data statistical filters were applied based on the relative standard deviations. The suggested threshold was 30%, followed by further normalization using log transformation (base 10) that applies a mathematical transformation to individual samples to adjust systematic differences among samples, “mean-centred and divided by the square root of the standard deviation of each variable” denoted as Pareto scaling was applied to adjust each metabolite in the sample by the scaling factor computed based on the dispersion of the variable. To test sample variability, principal component analysis (PCA) results were generated in a score plot displaying a 95% confidence region to distinguish variance in the different treatments. Partial least squares discriminant analysis (PLS-DA) was performed for datasets that did not show clear clustering, and the various sample treatments were successfully clustered. To visualize the separations with the relative concentrations of the corresponding metabolites in each sample, the top 15 features were shown on a variable importance projection (VIP) scores plot calculated based on a weighted sum of absolute regression.

## Figures and Tables

**Figure 1 plants-14-00531-f001:**
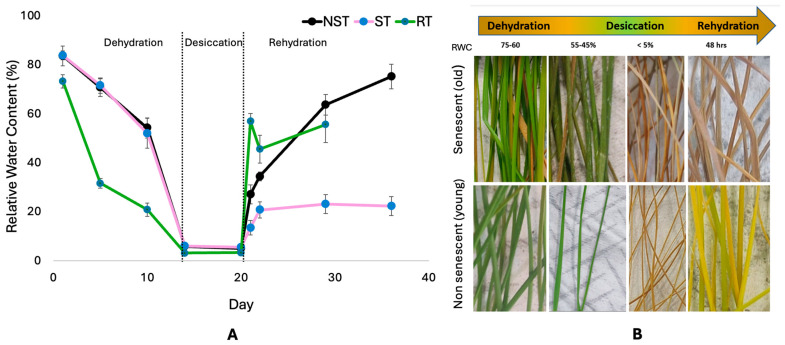
Changes in relative water content (RWC) of *E. nindensis* tissues used for sampling. Five biological replicates were used for the experiments, and error bars represent the standard errors across the biological replicates. (**A**) The graph shows the relative water content of NST, ST, and roots during dehydration, desiccation and rehydration time course. (**B**) shows visual changes in the NST and ST during the dehydration and rehydration time course, with only the NST regaining pigment by 48 h post-rehydration.

**Figure 2 plants-14-00531-f002:**
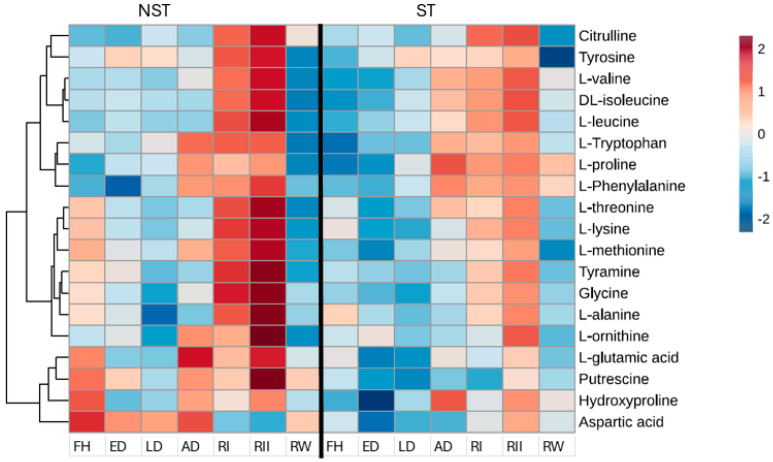
Heatmap of differentially abundant amino acids for non-senescent (NST) and senescent (ST) tissue in *E. nindensis*. The dehydration stages represented are an early response to dehydration (ED), late response to dehydration (LD), and air-dry (AD), 24 h post-rehydration (RI), 48 h post-rehydration (RII), and 1−week post-rehydration (RW) to fully hydrated (FH). Shades of maroon represent a high abundance of metabolites, and shades of blue represent a low abundance of metabolites.

**Figure 3 plants-14-00531-f003:**
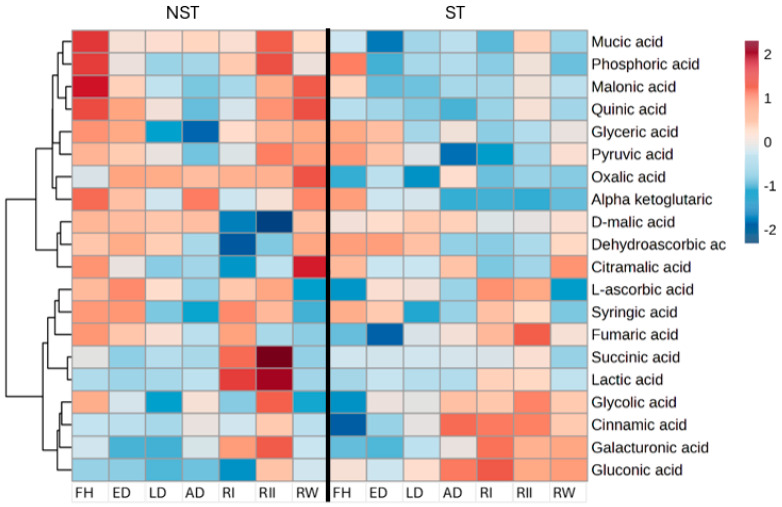
Heatmap of differentially abundant organic acids for non-senescent (NST) and senescent (ST) tissue in *E. nindensis*. The dehydration stages represented are an early response to dehydration (ED), late response to dehydration (LD), and air-dry (AD), 24 h post-rehydration (RI), 48 h post-rehydration (RII), and 1−week post-rehydration (RW) to fully hydrated (FH). Shades of maroon represent a high abundance of metabolites, and shades of blue represent a low abundance of metabolites.

**Figure 4 plants-14-00531-f004:**
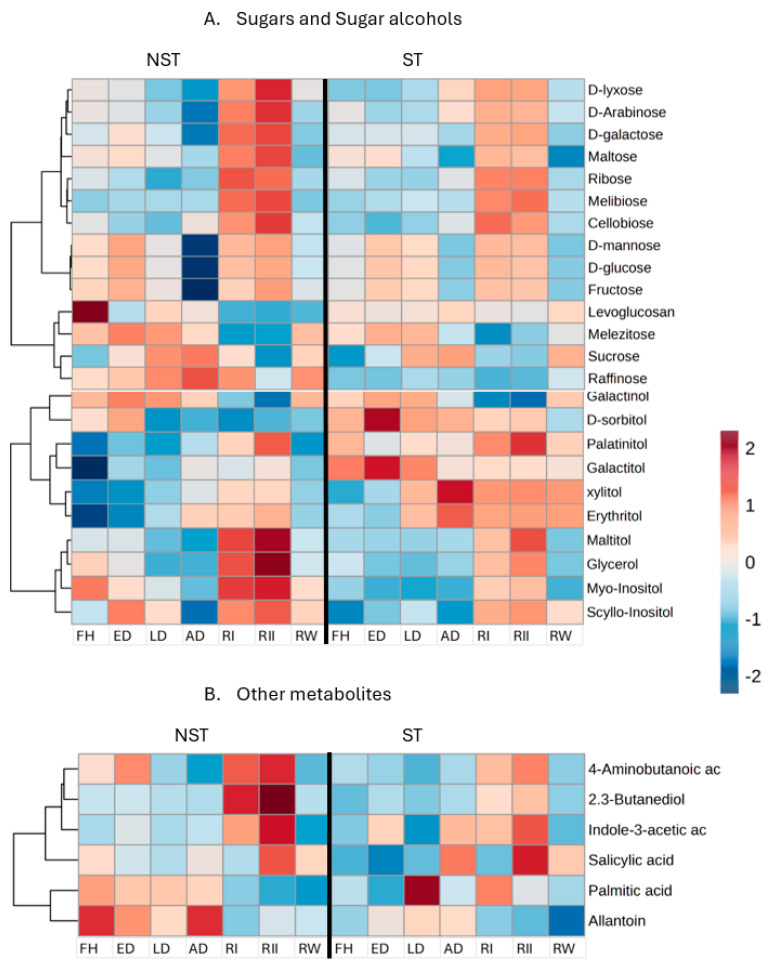
Heatmap of differentially abundant metabolites (**A**) sugars and sugar alcohols and (**B**) other metabolites among non-senescent (NST) and senescent (ST) tissue in *E. nindensis*. The dehydration stages represented are an early response to dehydration (ED), late response to dehydration (LD), and air-dry (AD), 24 h post-rehydration (RI), 48 h post-rehydration (RII), and 1−week post-rehydration (RW) to fully hydrated (FH). Shades of maroon represent a high abundance of metabolites, and shades of blue represent a low abundance of metabolites.

**Figure 5 plants-14-00531-f005:**
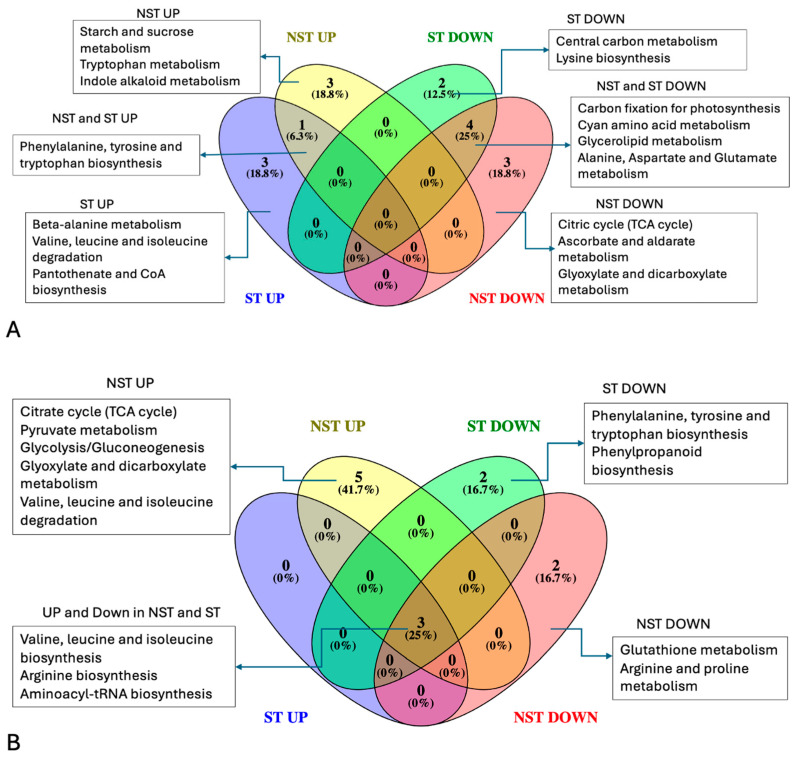
Venn diagram of metabolic pathways associated with differentially abundant metabolites in NST and ST of *E. nindensis* during dehydration and rehydration relative to fully hydrated and air-dry tissue, respectively. Venn diagram (**A**) indicates dehydration-associated metabolic pathways, and Venn diagram (**B**) indicates rehydration-related metabolic pathways. “UP” refers to metabolic pathways related to highly abundant metabolites, and “DOWN” is associated with diminishing metabolites. All metabolic pathways analysis were performed with MetaboAnalyst 5.0 using *A. thaliana* as the pathway library with a *p*-value of 0.05. Central carbon metabolism = glycolysis or gluconeogenesis.

**Figure 6 plants-14-00531-f006:**
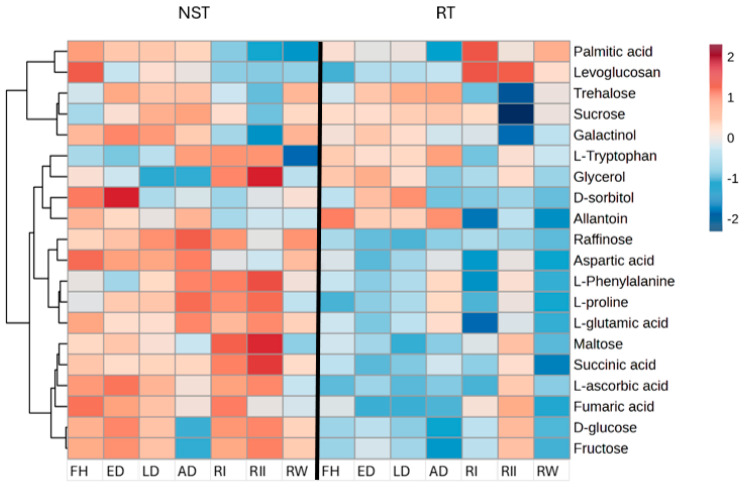
Heatmap of differentially abundant metabolites’ sugars, amino acids, organic acids, and sugar alcohols for non-senescent (NST) and root (RT) tissue in *E. nindensis*. The dehydration stages represented are an early response to dehydration (ED), late response to dehydration (LD), and air-dry (AD), 24 h post-rehydration (RI), 48 h post-rehydration (RII), and 1−week post-rehydration (RW) to fully hydrated (FH). Shades of maroon represent a high abundance of metabolites and shades.

## Data Availability

Data used in this study are available in the Appendix A.

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
