# Peer review of "Metabolite Profiling of the Resurrection Grass Eragrostis nindensis During Desiccation and Recovery"

_plants, 2025, doi:10.3390/plants14040531_

Round 1
Reviewer 1 Report
Comments and Suggestions for Authors
Metabolite profiling of the resurrection grass Eragrostis nindensis during desiccation and recovery. By Baluku et al.
In this manuscript, the authors focus on plant metabolite changes during desiccation and recovery.
I read this short article with interest, even if the results concern a specific (for me) group of plants.
There are practically no critical notes in this manuscript. The proportions between the introduction, results, and discussion are well-balanced. Apart from minor comments below, the methods are also well described.
Regarding Figure 5, I have a question for the authors. Why are glycolysis and gluconeogenesis among the processes in the cell that are either stimulated or inhibited under certain physiological conditions? I am asking in the context of the regulation mechanism of these two processes, which are considered opposing. Due to the similarity with other metabolic pathways, choosing a specific process, glycolysis or gluconeogenesis, might be worthwhile.
Minor comments:
line 118 please write “amino acidsamong” correctly
line 457-458 please add the temperature for centrifugation
line 132, line 152, line 176, line 214 Figure 2, Figure 3, Figure 4, Figure 6: please add RII
Figure 5: if possible, please enlarge the fonts for numbers and % values in the diagram
This manuscript can be published with minor changes.
Author Response
Dear Editor,
Thank you for allowing us to resubmit our manuscript entitled “Metabolite profiling of the resurrection grass Eragrostis nindensis during desiccation and recovery” to this special edition of Plants.
In response to the reviewers comments, we thank them for the positive statements and below I address their queries and concerns.
Our responses are in bold. All corrections made in the text of the paper are in red font.
REVIEWER 1
Regarding Figure 5, I have a question for the authors. Why are glycolysis and gluconeogenesis among the processes in the cell that are either stimulated or inhibited under certain physiological conditions? I am asking in the context of the regulation mechanism of these two processes, which are considered opposing. Due to the similarity with other metabolic pathways, choosing a specific process, glycolysis or gluconeogenesis, might be worthwhile.
Due to software limitations (which simply combines the processes) and given the GC MS standards we used, and limits of detection and quantification, we cannot determine for certain which process is favoured. We can speculate, based on the fact our data shows shutdown of the TCA cycle during dehydration, that glycolysis is favoured. However, there is an increase in glucose during drying and this could suggest gluconeogenesis. Without data on ATP/ADP turnover and respiration this is again speculation. We chose to not speculate overly and labelled the processes Central Carbon Metabolism.
Minor comments:
line 118 please write “amino acidsamong” correctly
line 457-458 please add the temperature for centrifugation
line 132, line 152, line 176, line 214 Figure 2, Figure 3, Figure 4, Figure 6: please add RII
Figure 5: if possible, please enlarge the fonts for numbers and % values in the diagram
All the above have been attended to.
Reviewer 2 Report
Comments and Suggestions for Authors
The study concerns an important subject of plant resistance to dehydration conditions, crucial in the context of climate changes. The manuscript contains valuable results and is well prepared.
The background is clearly written, all necessary issues are characterized, the choice of experimental model is explained. The aim of the study is suitably presented and justified.
The methods were properly selected and described.
The results are clearly presented, the figures are relevant and understandable.
The discussion is interesting and based on existing knowledge. Conclusions are relevant, however, the last sentence is not understandable.
Lines 418-419. “Therefore, this study will complement recently completed transcriptomic studies on this phenomenon [75], which collectively will provide further insight as to how we can improve drought tolerance, without yield penalty, in crops such as Teff.” The bibliographic source [75] is not completed, with only year (2019) - of publication? study? If these results were not published, they cannot be cited! And the entire sentence should be changed or deleted.
Other comments:
Line 28. „Plants have evolved numerous mechanisms to avoid, resist, or escape water deficit under drought conditions.” The term “escape” is not easily acceptable in case of plants (as immobile organisms), so please – either explain what you exactly mean, or delete this word.
Line 105. Please finish the sentence with a dot.
Line 118. Please delete “Amino acids” at the beginning of the paragraph.
Author Response
Dear Editor,
Thank you for allowing us to resubmit our manuscript entitled “Metabolite profiling of the resurrection grass Eragrostis nindensis during desiccation and recovery” to this special edition of Plants.
In response to the reviewers comments, we thank them for the positive statements and below I address their queries and concerns.
Our responses in bold. All corrections made in the text of the paper are in red font.
REVIEWER 2
Lines 418-419. “Therefore, this study will complement recently completed transcriptomic studies on this phenomenon [75], which collectively will provide further insight as to how we can improve drought tolerance, without yield penalty, in crops such as Teff.” The bibliographic source [75] is not completed, with only year (2019) - of publication? study? If these results were not published, they cannot be cited! And the entire sentence should be changed or deleted.
This sentence and reference have been deleted.
Other comments:
Line 28. „Plants have evolved numerous mechanisms to avoid, resist, or escape water deficit under drought conditions.” The term “escape” is not easily acceptable in case of plants (as immobile organisms), so please – either explain what you exactly mean, or delete this word.
These 3 terms are frequently used in the literature, with “escape” referring to the strategy of annuals where the adult dies, but desiccated seeds remain in the environment when conditions are not conductive to plant growth. E.g. see Baso et al., 2016 https://doi.org/10.12688/f1000research.7678.1. We thus have retained the wording.
Line 105. Please finish the sentence with a dot.
Done
Line 118. Please delete “Amino acids” at the beginning of the paragraph.
Done
Reviewer 3 Report
Comments and Suggestions for Authors
Eragrostis nindensis is a desiccation-tolerant grass. In this study, authors identified differentially abundant metabolites in leaves of senescent and non-senescent and root systems and found that raffinose, sucrose, glutamic acid, aspartic acid, proline, alpha-ketoglutaric acid, and allantoin act as major drivers for plant desiccation tolerance and aid the plant post-rehydration. This work gives further insight into the whole plant responses of E. nindensis to extreme dehydration conditions. However, on the whole, the research content and analysis are relatively simple, and there are many mistakes in the author's writing.
1. Line 439 “six plants were sampled for each treatment” the sample size seems small.
2. Line 442 ”(below)” What do you mean?
3. How many biological replicates are used for metabolite analysis?
4. Line 113 “of which 83 were differentially abundant” It seems that 51 metabolites are differentially abundant.
5. In this study, all the heatmaps were constructed by normalization using log10 transformation. The results do not provide information on which two samples are significantly different. Thus, I suggest that the authors add information about “Fold change” of differentially abundant metabolites according to peak area during the dehydration and rehydration compared to FH treatment in Supplementary data or Table.
This allows the reader to better understand the process of metabolite accumulation
6. Please add full name of RII for Figures 2-4, and 6.
7. Figure 6 “sugar alcohols and (B) for non-senescent” please delete (B) in line 212.
8. “Amino acids Among the amino acids”, please delete “Amino acids” in line 118
9. Supplementary data, A2 not be showed in the manuscript.
10. Figure 4 “Heatmap of differentially abundant metabolites (A) sugars and sugar alcohols and (B) nonsenescent (NST) and senescent (ST) tissue in E. nindensis.” in lines 173-174. (B) should be other metabolites.
Author Response
Dear Editor,
Thank you for allowing us to resubmit our manuscript entitled “Metabolite profiling of the resurrection grass Eragrostis nindensis during desiccation and recovery” to this special edition of Plants.
In response to the reviewers comments, we thank them for the positive statements and below I address their queries and concerns.
Our responses in bold. All corrections made in the text of the paper are in red font.
REVIEWER 3
1. Line 439 “six plants were sampled for each treatment” the sample size seems small.
It is a natural consequence that in use of field collected resurrection plants, numbers of plants permitted to be collected is restricted by collection permits and given the slow growing and long life cycles of many of these plants, seed collection is slow. Both prevent large number of plants for destructive sampling at any one time. This is unlike the situation in use of Arabidopsis or crop plants, so reviewers working on such plants are unaware of our unavoidable situation. I can cite many papers, not only my own, attesting to this fact. Having said that, not only 6 plants were used, but from a bigger population, at each sampling, 6 different individuals were harvested. In order to prevent additional stress, no one plant was sampled frequently. This methodology section has been rewritten to clarify this.
2. Line 442 ”(below)” What do you mean?
3. How many biological replicates are used for metabolite analysis?
Both 2 and 3 are addressed in our re-wording of the methods section pertaining to these queries.
4. Line 113 “of which 83 were differentially abundant” It seems that 51 metabolites are differentially abundant.
We identified 97 metabolites of which 83 were differentially abundant. This is shown in Supplementary Table A1
5. In this study, all the heatmaps were constructed by normalization using log10 transformation. The results do not provide information on which two samples are significantly different. Thus, I suggest that the authors add information about “Fold change” of differentially abundant metabolites according to peak area during the dehydration and rehydration compared to FH treatment in Supplementary data or Table. This allows the reader to better understand the process of metabolite accumulation
We have provided this information in an Excel spreadsheet, now in the Supplementary data.
6. Please add full name of RII for Figures 2-4, and 6.
7. Figure 6 “sugar alcohols and (B) for non-senescent” please delete (B) in line 212.
8. “Amino acids Among the amino acids”, please delete “Amino acids” in line 118
Points 6-8 were done.
9. Supplementary data, A2 not be showed in the manuscript.
Thank you for pointing this out. We have now included reference to it.
10. Figure 4 “Heatmap of differentially abundant metabolites (A) sugars and sugar alcohols and (B) nonsenescent (NST) and senescent (ST) tissue in E. nindensis.” in lines 173-174. (B) should be other metabolites.
Done.
Round 2
Reviewer 3 Report
Comments and Suggestions for Authors
Eragrostis nindensis is a desiccation-tolerant grass. In this study, authors identified differentially abundant metabolites in leaves of senescent and non-senescent and root systems and found that raffinose, sucrose, glutamic acid, aspartic acid, proline, alpha-ketoglutaric acid, and allantoin act as major drivers for plant desiccation tolerance and aid the plant post-rehydration. This work gives further insight into the whole plant responses of E. nindensis to extreme dehydration conditions. Now, manuscript is acceptable.